# INTERFACEDIFF: INTERFACE-AWARE SEQUENCE-STRUCTURE CO-DESIGN OF PROTEIN COMPLEXES WITH GRAPH-BASED DIFFUSION

## ABSTRACT

The rational design of protein–protein complexes remains a fundamental challenge in synthetic biology and therapeutic development. Current generative methods often fall short in performing sequence–structure co-design, particularly in treating the functionally critical protein-protein interface as a first-class target. To bridge this gap, we present InterfaceDiff, a graph-based diffusion framework for interface-aware co-design of protein complexes. The complex is encoded by intra-chain graphs coupled through an explicit bipartite interface graph, concentrating modeling capacity on physically interacting residues. InterfaceDiff learns a joint distribution over discrete amino acid sequences and continuous local rigid frames (rotations and translations) by a simultaneous denoising process. To achieve this efficiently, we develop a novel graph neural network denoiser inspired by Invariant Point Attention, which performs message passing on the sparse graph representation while avoiding the computational overhead of fully SE(3)-equivariant networks. We evaluate InterfaceDiff across multiple design tasks, demonstrating its ability to generate diverse, high-quality, and physically plausible all-atom complexes. Our method achieves strong performance on key biophysical and geometric metrics, offering a scalable and geometrically efficient approach for controllable protein complex engineering. This work establishes a foundation for generative co-design of novel molecular interactions.

## 1 INTRODUCTION

Protein–protein complexes (PPCs) orchestrate signal transduction, immune recognition, enzymatic regulation, and supramolecular assembly. The ability to rationally design new complexes would unlock powerful capabilities in synthetic biology and therapeutics (Aloy & Russell, 2002; Huang et al., 2016). At the heart of PPC design lies the *protein–protein interface*: a compact, chemically heterogeneous region where cross-chain residues engage through hydrogen bonds, salt bridges, hydrophobic interactions, and so forth (Fig. 1). Treating the interface as a *first-class design target*, rather than a byproduct of global modeling, is essential for controlling binding affinity, specificity, and functional reprogramming (Kortemme & Baker, 2004). Practical design systems must also navigate the dual discreteness and continuity of proteins: amino-acid sequences are discrete, whereas $3D$ conformations evolve continuously under local stereochemistry and global packing constraints.

Deep learning has transformed protein *folding* and *inverse folding*. Protein folding models (Jumper et al., 2021; Baek et al., 2021; Abramson et al., 2024; Boitreaud et al., 2024) attain high accuracy in predicting the structure of monomers and multi-chain assemblies and provide confidence estimates that have become standard currencies for evaluation. Inverse folding approaches (Dauparas et al., 2022; Yi et al., 2023; Zhu et al., 2024) design novel sequences based on *fixed* backbones of native protein. Concurrently, diffusion models have further advanced researches related to protein (Watson et al., 2023; Yim et al., 2023; Ingraham et al., 2023). These advances supply powerful building blocks for design, yet they typically optimize *either* structure *or* sequence in isolation: sequences are tuned post hoc against a frozen backbone, or structures are predicted for given sequences without exploring sequence–structure co-variation.

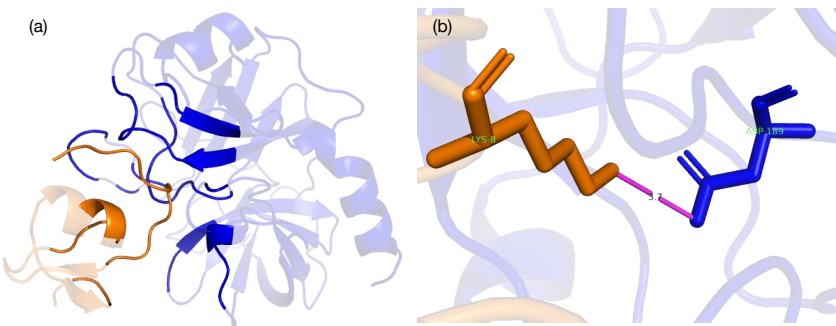

Figure 1: Illustraion of protein-protein interface. (a) Crystal structure of a protein-protein complex (PDB ID: 1AN1). The interface region is highlighted in opaque representation (blue and orange), while the remaining non-interface regions are shown in a semi-transparent view. (b) Close-up view of the interface in 1AN1. The figure shows the salt bridge interaction between Lysine (LYS) in chain I (orange) and Aspartic Acid (ASP) in chain E (blue), with a distance of 3.7 Å marked.

Previous studies on protein complex design roughly fall into one of the two categories: (i) focusing only on a specific component of the complex—for instance, binder design targets only the binder, and antibody design focuses exclusively on the CDRs; (ii) generating complexes of random lengths, where the plausibility of the results is difficult to establish without experimental validation, even if they perform well under their chosen evaluation metrics. In contrast to these paradigms, we pursue *sequence–structure co-design* of entire protein complexes grounded in real data, with an explicit emphasis on the *interface*.

Despite rapid progress, a concrete gap remains for *interface-aware co-design*. First, complex *prediction* systems (Abramson et al., 2024; Boitreaud et al., 2024) predominantly infer conformations for *given* sequences, rather than *designing* new sequences and geometries in tandem. Second, many design pipelines assume rigid or near-rigid backbones (e.g., inverse folding on fixed scaffolds or rigid-body docking), leaving interface remodeling largely implicit and providing limited control over contact patterns and energetics (Hsu et al., 2022; Desta et al., 2020; Harmalkar & Gray, 2021). Third, fully SE(3)-equivariant architectures, while expressive, often incur substantial memory and engineering overheads that complicate scaling to large or sparse multi-chain graphs (Satorras et al., 2021; Wang et al., 2024). Together, these limitations hinder *controllable* and *diverse* interface design where mutations, local rearrangements, and cross-chain contacts must co-evolve.

Interfaces are naturally represented as graphs: intra-chain neighborhoods capture local stereochemistry, while cross-chain contacts define the interface topology (Pancino et al., 2024; Xu et al., 2024). However, most generative pipelines either omit an explicit interface representation or fuse it into dense all-to-all attention, blurring the boundary between intra-chain modeling and cross-chain design. An explicit *interface graph* (Si & Yan, 2024) can disentangle these roles. It focuses modeling capacity on the residues that determine binding, exposes controllable knobs, and eases scaling via sparse message passing. Moreover, operating in residue-centered local frames provides geometric robustness to global rigid-body motion while preserving fine-grained orientation cues (Jumper et al., 2021; Randolph & Kuhlman, 2024).

These observations above yield four core design principles for effective interfaces.

(1) **Interface awareness.** Elevate the cross-chain interface to a primary generative target via an explicit representation (e.g., a bipartite contact graph) rather than relying on it to emerge implicitly (Si & Yan, 2024; Gainza et al., 2020).

(2) **Joint co-design.** Model sequence and 3D geometry *jointly*, allowing mutations and conformational changes to co-evolve during generation instead of being optimized in disjoint stages (Ingraham et al., 2023; Hayes et al., 2025).

(3) **Geometric efficiency.** Capture local rigid-body cues with residue-centered frames while avoiding the computational burden of heavy fully equivariant stacks, enabling sparse message passing on large and heterogeneous complexes (Luo et al., 2024; Xie et al., 2024).

(4) **Full-atom readiness.** Convert backbone-level designs to all-atom structures for physical validation and downstream use (e.g., sterics, packing, and rotameric checks) (McPartlon & Xu, 2023; Lee & Kim, 2025).

We address these needs by introducing *InterfaceDiff*, an interface-aware, graph-based diffusion framework for *joint* sequence–structure design of protein-protein complexes. At a high level, we encode each chain as a residue graph and extract an explicit *interface bipartite graph* by connecting cross-chain residues within a C$\alpha$-distance threshold. We then co-generate discrete sequences and local rigid frames defined by residue-centered coordinates (N–C$\alpha$–C), using a lightweight denoiser inspired by invariant point attention to perform sparse message passing over intra-chain neighborhoods and the interface graph. Finally, sampled backbones are converted to full-atom models via a packing step to enable standard structural checks. This interface-centric formulation offers controllability and diversity to complex design, while remaining computationally efficient.

In summary, our paper makes the following contributions:

- We propose *InterfaceDiff*, an interface-aware, graph-based diffusion framework for *joint* sequence–structure design of protein–protein complexes. Each chain is encoded as a residue graph, and an explicit *interface bipartite graph* connects cross-chain residues; the model co-generates discrete sequences and local rigid frames.
- Our model employs a geometrically efficient, IPA-inspired sparse denoiser operating in residue-local frames to capture 3D cues without heavy fully SE(3)-equivariant tensors, improving scalability to large and sparse complexes.
- The framework applies to sequence-structure co-design, fixed-backbone inverse folding with structure prediction, binder design, and antibody design, and integrates seamlessly with all-atom packing for practical use. Comprehensive experiments show consistent improvements over baselines while producing diverse, physically plausible interfaces.

## 2 METHODS

In this section, we present **InterfaceDiff**, an interface-aware diffusion framework for joint sequence–structure design of protein–protein complexes. Section 2.1 defines the residue-level graph representation of each chain together with the bipartite interface graph. Section 2.2 formalizes the design problem and outlines the overall framework. Section 2.3 specifies the coupled diffusion over amino-acid identities and residue-local frames. Section 2.4 details the denoising network parameterization. Finally, Section 2.5 presents the sampling algorithms for generating diverse, physically plausible complexes.

### 2.1 RESIDUE- AND INTERFACE-LEVEL GRAPHS.

A protein–protein complex contains multiple chains; chain $c$ comprises an ordered set of amino-acid (AA) residues $\mathcal{V}_c$ with $N_c = |\mathcal{V}_c|$. For each chain we construct a residue-level graph

$$\mathcal{G}_c^{\text{chain}} = \left( X_c^{\text{chain}}, \, A_c^{\text{chain}}, \, E_c^{\text{chain}} \right).$$

Every node $i \in \mathcal{V}_c$ represents one residue with attributes

$$x_i = \left[ x_i^{\text{aa}}, \, R_i, \, x_i^{\text{trans}}, \, x_i^{\text{prop}} \right],$$

where $x_i^{\text{aa}} \in \mathbb{R}^{20}$ is a one-hot AA type, $R_i \in \text{SO}(3)$ is the local coordinate frame constructed from backbone atoms $(N, C_\alpha, C)$, $x_i^{\text{trans}} \in \mathbb{R}^3$ is the local-frame origin given by the global coordinate of $C_\alpha$, and $x_i^{\text{prop}}$ are physicochemical features. Edges in $A_c^{\text{chain}}$ are built by a $k$-nearest-neighbour (kNN) rule in $C_\alpha$ space, with $k$ chosen as a function of chain length; $E_c^{\text{chain}}$ encodes sequence-aware and geometric features (details in Appendix A.2). Working in residue-level local frames $(R_i, x_i^{\text{trans}})$ preserves geometry while avoiding heavy global SE(3)-equivariant denoisers.

To capture cross-chain contacts, we build a bipartite interface graph

$$\mathcal{G}^{\text{inter}} = \left( X^{\text{inter}}, A^{\text{inter}}, E^{\text{inter}} \right),$$

whose nodes are residues belonging to different chains and whose attributes mirror those above. An interface edge is added between residues from distinct chains if their minimum inter-atomic distance is below a threshold $\tau$. $E^{\text{inter}}$ aggregates geometry and interaction semantics (hydrogen bonds, salt bridges, hydrophobics, and chemical complementarity indicators), yielding a fine-grained interface representation.

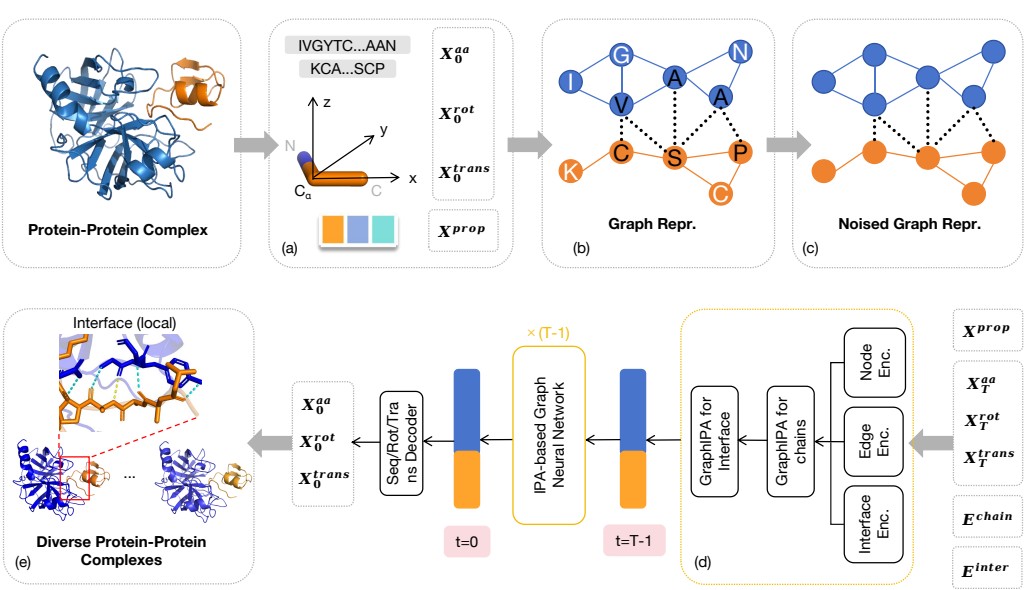

Figure 2: **Overview of *InterfaceDiff*.** Top: diffusion process; bottom: denoising process. (a) Extract sequence, structure, and chemical features. (b) Build intra-chain graphs with kNN (solid edges) and inter-chain graphs via distance thresholds (dashed edges), with rich edge features. (c) Add noise to sequence, rotation, and translation to obtain a noised graph. (d) Pass the noised graph and time embeddings into the denoising network, updating node features, which MLP decoders map to sequence, rotation, and translation. (e) Post-processing with side-chain packing reconstructs complete complexes, enabling diverse sequences and structures via multiple samples.

## 2.2 PROBLEM FORMULATION

Let the complex-level graph be
$$\mathcal{G} := \big(\{\mathcal{G}_c^{\text{chain}}\}_c, \, \mathcal{G}^{\text{inter}}\big).$$
Denote $X^{\text{aa}} = \{x_i^{\text{aa}}\}_{i=1}^N$, $X^{\text{rot}} = \{R_i\}_{i=1}^N$ with $R_i \in \text{SO}(3)$, and $X^{\text{trans}} = \{x_i^{\text{trans}}\}_{i=1}^N$ where $N = \sum_c N_c$. We aim to model the conditional density
$$p_\theta\big(X^{\text{aa}}, X^{\text{rot}}, X^{\text{trans}} \mid \mathcal{G}\big),$$
and sample diverse yet physically consistent complexes.

We instantiate a discrete–continuous diffusion over $X = \{(x_i^{\text{aa}}, R_i, x_i^{\text{trans}})\}_{i=1}^N$. A forward process corrupts $X_0 \to X_T$, while a reverse kernel
$$p_\theta(X_{t-1} \mid X_t, \mathcal{G})$$
is parameterized by an IPA-based graph neural network attending jointly to $(E_c^{\text{chain}})_c$ and $E^{\text{inter}}$. At inference time, ancestral sampling from $p_\theta$ produces $(\hat{X}^{\text{aa}}, \hat{X}^{\text{rot}}, \hat{X}^{\text{trans}})$ (Fig. 2).

## 2.3 DIFFUSION PROCESSES

We employ the cosine variance schedule (Nichol & Dhariwal, 2021) with offset $s = 0.008$:

$$\bar{\alpha}_t = \frac{\cos^2\big(\frac{\pi}{2}\frac{t/T+s}{1+s}\big)}{\cos^2\big(\frac{\pi}{2}\frac{s}{1+s}\big)}, \qquad \alpha_t = 1 - \beta_t, \qquad \beta_t = 1 - \frac{\bar{\alpha}_t}{\bar{\alpha}_{t-1}}. \tag{1}$$

**Rotations on** $\text{SO}(3)$ **via axis–angle with noise prediction.** Let $[\cdot]_\times : \mathbb{R}^3 \to \mathfrak{so}(3)$ be the skew-symmetric operator and $\exp : \mathfrak{so}(3) \to \text{SO}(3)$ the matrix exponential.[1] For residue $i$, define the

---
[1] We use a numerically stable Rodrigues formulation for $\exp([\cdot]_\times)$ with small-angle series.

clean axis–angle vector $\mathbf{v}_{0,i} \in \mathbb{R}^3$ from its clean frame $R_{0,i}$ by $R_{0,i} = \exp([\mathbf{v}_{0,i}]_\times)$. The forward process adds Gaussian noise in axis–angle space:

$$\mathbf{v}_{t,i} = \sqrt{\bar{\alpha}_t}\,\mathbf{v}_{0,i} + \sqrt{1-\bar{\alpha}_t}\,\boldsymbol{\epsilon}_{t,i}^{\text{so3}}, \quad \boldsymbol{\epsilon}_{t,i}^{\text{so3}} \sim \mathcal{N}(\mathbf{0}, \mathbf{I}_3), \quad R_{t,i} = \exp([\mathbf{v}_{t,i}]_\times). \tag{2}$$

Given the noised $R_{t,i}$, the denoiser predicts the *so(3) noise* $\widehat{\boldsymbol{\delta}}_{t,i} \in \mathbb{R}^3$ from which we form an incremental rotation $U_{t,i} = \exp([\widehat{\boldsymbol{\delta}}_{t,i}]_\times)$. We update by right-multiplication[2]:

$$\widehat{R}_{0|t,i} = R_{t,i}\,U_{t,i}. \tag{3}$$

The rotation loss compares the updated frame to the clean frame using a trace form

$$\ell_{\text{rot}}(i) = 3 - \text{tr}\big(\widehat{R}_{0|t,i}^\top R_{0,i}\big). \tag{4}$$

**Translations in $\mathbb{R}^3$.** Let $\tilde{x}_{0,i}^{\text{trans}}$ be standardized coordinates in the local frame. The forward diffusion is the standard DDPM:

$$\tilde{x}_{t,i}^{\text{trans}} = \sqrt{\bar{\alpha}_t}\,\tilde{x}_{0,i}^{\text{trans}} + \sqrt{1-\bar{\alpha}_t}\,\boldsymbol{\epsilon}_{t,i}^{\text{trans}}, \qquad \boldsymbol{\epsilon}_{t,i}^{\text{trans}} \sim \mathcal{N}(\mathbf{0}, \mathbf{I}_3). \tag{5}$$

The network predicts local-frame noise $\widehat{\epsilon}_{t,i}^{\text{local}}$, which is rotated to the global frame by the current orientation: $\widehat{\epsilon}_{t,i} = R_{t,i}\,\widehat{\epsilon}_{t,i}^{\text{local}}$. We use the per-residue noise MSE $\ell_{\text{pos}}(i) = \|\widehat{\epsilon}_{t,i} - \boldsymbol{\epsilon}_{t,i}^{\text{trans}}\|_2^2$, and the standard reverse mean update:

$$\tilde{x}_{t-1,i}^{\text{trans}} = \frac{1}{\sqrt{\alpha_t}}\left(\tilde{x}_{t,i}^{\text{trans}} - \frac{1-\alpha_t}{\sqrt{1-\bar{\alpha}_t}}\,\widehat{\epsilon}_{t,i}\right) + \sigma_t\,\mathbf{z}, \quad \mathbf{z} \sim \mathcal{N}(\mathbf{0}, \mathbf{I}_3). \tag{6}$$

**Amino-acid types (discrete channel).** Let $K = 20$ and $\mathbf{c}_{t,i} \in \Delta^{K-1}$ be type probabilities with $x_{t,i}^{\text{aa}} \sim \text{Cat}(\mathbf{c}_{t,i})$. Two diffusion channels can be applied:

*(i) Uniform multinomial.*

$$q(\mathbf{c}_{t,i} \mid \mathbf{c}_{0,i}) = \bar{\alpha}_t\,\mathbf{c}_{0,i} + (1-\bar{\alpha}_t)\tfrac{1}{K}\mathbf{1}, \quad x_{t,i}^{\text{aa}} \sim \text{Cat}(\mathbf{c}_{t,i}), \tag{7}$$

with the exact posterior at $t-1$ (given the denoiser's $\widehat{\mathbf{c}}_{0,i}$)

$$\boldsymbol{\theta}_{t,i} \propto \big(\alpha_t\,\mathbf{c}_{t,i} + (1-\alpha_t)\tfrac{1}{K}\mathbf{1}\big) \odot \big(\bar{\alpha}_{t-1}\,\widehat{\mathbf{c}}_{0,i} + (1-\bar{\alpha}_{t-1})\tfrac{1}{K}\mathbf{1}\big), \tag{8}$$

and KL loss $\ell_{\text{seq}}(i) = \text{KL}\big(q \,\|\, q_\theta\big)$.

*(ii) BLOSUM-guided Markov kernel.* Let $\mathbf{Q}_t \in \mathbb{R}^{K \times K}$ be a temperature-annealed transition from BLOSUM62 and $\bar{\mathbf{Q}}_t = \prod_{\tau=1}^{t} \mathbf{Q}_\tau$ (Yi et al., 2023). Then

$$\mathbf{c}_{t,i} = \bar{\mathbf{Q}}_t\,\mathbf{c}_{0,i}, \quad x_{t,i}^{\text{aa}} \sim \text{Cat}(\mathbf{c}_{t,i}), \tag{9}$$

with exact posterior

$$\boldsymbol{\theta}_{t,i} = \frac{\mathbf{e}_{t,i}\mathbf{Q}_t^\top \odot \big(\widehat{\mathbf{c}}_{0,i}\bar{\mathbf{Q}}_{t-1}\big)}{\big(\widehat{\mathbf{c}}_{0,i}\bar{\mathbf{Q}}_t\big)\,\mathbf{e}_{t,i}^\top}, \quad \mathbf{e}_{t,i} = \text{onehot}(x_{t,i}^{\text{aa}}), \tag{10}$$

and the same KL objective.

**Training objective (per-residue).** All channels share the schedule coefficients $\{\bar{\alpha}_t, \alpha_t, \beta_t\}$ and are denoised jointly by the shared IPA-based network conditioned on $\mathcal{G}$. We compute *per-residue* losses and sum over *all* residues:

$$\mathcal{L} = \sum_{i=1}^{N}\Big(\lambda_{\text{rot}}\ell_{\text{rot}}(i) + \lambda_{\text{pos}}\ell_{\text{pos}}(i) + \lambda_{\text{seq}}\ell_{\text{seq}}(i)\Big), \tag{11}$$

where $\lambda_{\text{rot}}, \lambda_{\text{pos}}, \lambda_{\text{seq}}$ are hyperparameters.

---

[2]Right-multiplication is consistent with treating $\widehat{\boldsymbol{\delta}}$ as a local-frame update. Left-multiplication is equivalent under a different convention.

Table 1: Performance comparison of in Sequence-Structure Co-Design tasks.

| Model | Sequence | | | Structure | | | Interface | | |
|---|---|---|---|---|---|---|---|---|---|
| | Recovery Rate%↑ | Perplexity↓ | Diversity↑ | TM-score↑ | RMSD↓ | lDDT↑ | Success Rate↑ | $E$↓ | $\Delta G$↓ |
| AlphaDesign | 39.92 | 7.85 | 0.19 | 0.74 | 17.75 | 0.79 | 0.16 | 33.85 | 20.38 |
| ESM-IF1 | 42.83 | 6.95 | 0.23 | 0.81 | 8.82 | 0.84 | 0.37 | -10.87 | -12.58 |
| ProteinMPNN | 48.37 | 5.46 | 0.28 | 0.87 | 6.81 | 0.88 | 0.46 | -23.54 | -18.49 |
| PiFold | 47.28 | 5.62 | 0.21 | 0.78 | 14.73 | 0.85 | 0.24 | 10.34 | 8.26 |
| GraDe-IF | 49.57 | 5.04 | 0.31 | 0.76 | 16.942 | 0.86 | 0.22 | 23.52 | 20.83 |
| Bridge-IF | 51.14 | **4.63** | 0.30 | 0.87 | 9.98 | 0.87 | 0.25 | 0.34 | -3.73 |
| InterfaceDiff | **53.08** | 4.87 | **0.32** | **0.88** | 3.47 | **0.89** | **0.56** | **-53.92** | **-47.33** |
| InterfaceDiff$_{bb}$ | 52.87 | 4.95 | 0.29 | 0.86 | **2.95** | 0.85 | 0.49 | -21.64 | -19.53 |

## 2.4 IPA-based Graph Denoiser Network

The reverse kernel $p_\theta(X_{t-1} \mid X_t, \mathcal{G})$ is parameterized by a shared graph neural network that processes both structural and chemical information. We first encode each chain graph $\mathcal{G}_c^{\text{chain}}$ independently, then initialize and process the interface graph $\mathcal{G}^{\text{inter}}$ to model cross-chain interactions. The core block is a geometric attention module (inspired by Invariant Point Attention from Jumper et al. (2021)) that updates residue features using edge relations $(E_c^{\text{chain}})_c$, $E^{\text{inter}}$ and SE(3)-invariant functions of the noisy frames $(R_{t,i}, x_{t,i}^{\text{trans}})$. After a series of these blocks, the final node representations are concatenated with a time embedding $t_{\text{embed}}$ and fed into three distinct decoders: (i) a sequence decoder producing $\widehat{c}_{0,i}$, (ii) a rotation decoder producing *so(3) noise* $\widehat{\delta}_{t,i}$ used in (3), and (iii) a translation decoder producing local-frame noise $\widehat{\epsilon}_{t,i}^{\text{local}}$ used in (5)–(6). Further details are in Appendix A.5.

## 2.5 Sampling Algorithms

At $t = T$ we initialize each residue $i$ independently:

$$R_{T,i} \sim \text{Unif}\big(\text{SO}(3)\big), \quad \tilde{x}_{T,i}^{\text{trans}} \sim \mathcal{N}(\mathbf{0}, \mathbf{I}_3), \quad x_{T,i}^{\text{aa}} \sim \text{Unif}\{1, \dots, 20\}.$$

Following this initialization, the model then runs a reverse process, iteratively updating $(x_{t,i}^{\text{aa}}, R_{t,i}, x_{t,i}^{\text{trans}}) \rightarrow (x_{t-1,i}^{\text{aa}}, R_{t-1,i}, x_{t-1,i}^{\text{trans}})$ using the decoders as above until $t = 0$. Finally, we reconstruct a full-atom structure in two steps. First, the backbone atoms $(N, C_\alpha, C, O)$ are assembled for each residue by applying idealized local geometries relative to their final predicted frames $R_{0,i}$ and $C_\alpha$ positions (Engh & Huber, 2006). Second, given the designed sequence and backbone, side chain atoms are placed with FlowPacker (Lee & Kim, 2025) to obtain the optimized all-atom $3D$ complex.

## 3 Experiments

In this section, we evaluate the proposed method through extensive experiments to demonstrate its effectiveness. The experiments were conducted on datasets derived from PDB(Burley et al., 2025), PDBbind v2020, and PINDER(Kovtun et al., 2024) via multi-stage filtering. We evaluate designed complexes from three dimensions—sequence, structure, and interface. Details of the filtering procedure and evaluation metrics are provided in Appendix A.6 and A.7.

### 3.1 Sequence-Structure Co-Design

We selected one of the most powerful inverse folding approaches developed in recent years and performed multiple rounds of sampling to obtain diverse sequence designs. These designed sequences were then used as input to Chai-1, a state-of-the-art model for biomolecular structure prediction, to generate structural predictions. Since most inverse folding models are designed for single-chain proteins, we concatenate single chains to design the full sequence of a protein complex.

We tuned prior inverse-folding baselines to their best sequence-recovery settings. As shown in Table 1, our method achieves higher recovery and intra-target diversity at competitive perplexity,

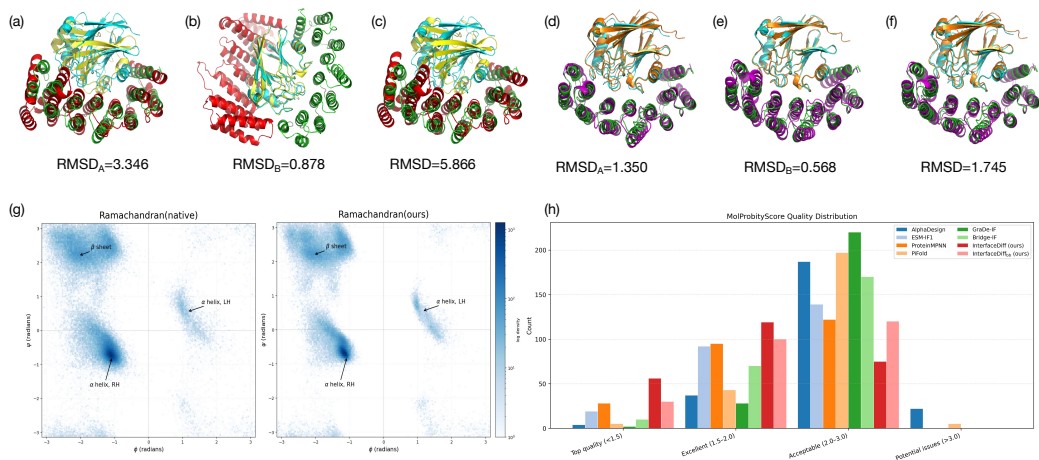

Figure 3: Visualization of sequence–structure co-design results. In each subpanel, green marks the native complex's long chain and cyan the native short chain. (a–c) GraDe-IF designs(red: designed long chain; yellow: designed short chain). (d–f) Our method (magenta: designed long chain; orange: designed short chain). The first column aligns the long chain, the second aligns the short chain, and the third aligns the entire complex. RMSD values are reported for the aligned components. (g) Ramachandran plot of the designed complexes. (h) Distribution of MolProbity Score.

and yields the most plausible backbones and side chains. Interface metrics further show the largest gains in cross-chain interactions. Notably, the low RMSDs reported for inverse-folding methods largely reflect single-chain cases; for complexes—where binding orientation and inter-chain contacts matter—their lack of explicit interface modeling degrades performance. On 2R17 (Fig. 3), GraDe-IF folds individual chains but misplaces the binding site and fails to assemble the complex, whereas our method correctly folds the entire complex.

### 3.2 Fixed backbone Sequence Design and Structure Prediction

Here we fix the backbone and sample sequences with our diffusion model, reusing the inverse-folding baselines from Table 1 for a controlled comparison. Unlike baselines that design sequences then fold, our pipeline *co-designs* sequences and structures: each denoising step predicts residue types conditioned on the backbone and cross-chain context, yielding a self-consistent complex. As shown in the last row of Table 1, our method matches or exceeds baselines in sequence recovery while maintaining higher diversity and competitive perplexity, and remains strong on structure-level metrics. Notably, fixing the backbone reduces RMSD yet does not uniformly improve interface-level quality. A rigid backbone restricts side-chain rotamers at the interface, often leaving buried unsatisfied polar atoms, minor steric clashes, or torsional strain. In contrast, when the backbone is allowed to move, coordinated packing can simultaneously adjust backbone and side chains thus reducing clashes, which explains improvements in complex-level metrics under flexible-backbone settings. Qualitatively(Fig. 3), baselines fold monomers correctly yet mis-register the interface, whereas our co-designed complex attains the intended docking pose with well-packed contacts.

### 3.3 Target-Protein Binder Complex Design

In this setting, we apply INTERFACEDIFF to directly design binders for a given target protein. As summarized in Table 2, recovery rate increases while perplexity decreases, and diversity is maintained or slightly improved, indicating that the model sharpens interface-specific preferences without collapsing to a single mode. The predicted conformations exhibit higher TM-score and lower backbone RMSD, suggesting closer agreement with target backbones. Besides, interface quality and energetics improve in tandem: lDDT rises, the total energy $E$ is lower, and the binding energy $\Delta G$ becomes more favorable. The case study(PDB ID:2R17) in Fig. 4 illustrates that although fixing one chain improves the inverse-folding model's sequence-level metrics and the designed binder attains a lower structural RMSD, the model still struggles to identify the correct interface and to produce

Table 2: Performance comparison of in Target-Protein Binder Complex Design tasks.

| Model | Sequence | | | Structure | | | Interface | | |
|---|---|---|---|---|---|---|---|---|---|
| | Recovery Rate%↑ | Perplexity↓ | Diversity↑ | TM-score↑ | RMSD↓ | lDDT↑ | Success Rate↑ | $E$↓ | $\Delta G$↓ |
| AlphaDesign | 40.68 | 7.43 | 0.21 | 0.78 | 16.45 | 0.85 | 0.22 | 23.85 | 19.47 |
| ESM-IF1 | 44.93 | 5.78 | 0.27 | 0.84 | 6.48 | 0.86 | 0.49 | -22.36 | -18.32 |
| ProteinMPNN | 51.37 | 4.76 | 0.32 | 0.87 | 5.98 | **0.90** | 0.52 | -30.59 | -24.16 |
| PiFold | 49.15 | 5.23 | 0.22 | 0.83 | 11.36 | 0.86 | 0.30 | -8.34 | -2.28 |
| GraDe-IF | 52.57 | 4.89 | 0.31 | 0.81 | 12.04 | 0.88 | 0.32 | -6.10 | -9.35 |
| Bridge-IF | 53.64 | **4.24** | 0.29 | 0.88 | 7.51 | 0.87 | 0.45 | -19.72 | -14.26 |
| InterfaceDiff | **55.26** | 4.25 | **0.34** | **0.91** | **3.23** | **0.90** | **0.56** | **-53.92** | **-47.33** |

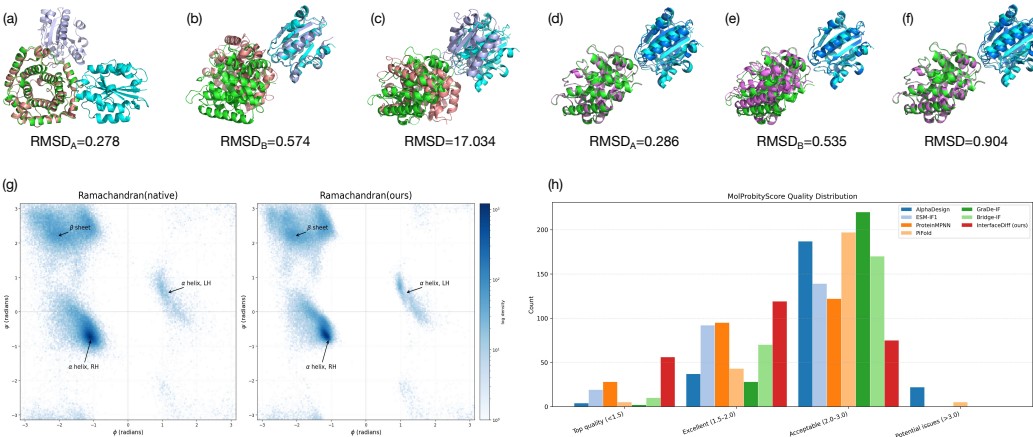

Figure 4: Visualization of Target-Protein Binder Design results. In each subpanel, green marks the native complex's long chain and cyan the native short chain. (a–c) ESM-IF1 designs (pink: Fixed long chain; purple: designed short chain). (d–f) Our method (magenta: Fixed long chain; blue: designed short chain). The first column aligns the long chain, the second aligns the short chain, and the third aligns the entire complex. RMSD values are reported for the aligned components. (g) Ramachandran plot of the designed complexes. (h) Distribution of MolProbity Score.

a structurally plausible complex, while our designed binder achieves smaller alignment error, and displays a consistent energetic funnel toward a physically plausible binding pose. INTERFACEDIFF jointly improves designability, foldability, and bindability without sacrificing sequence diversity, demonstrating effective absorption of interface constraints and cross-level consistency.

## 3.4 ABLATION STUDY

We conduct three ablations under the same training and evaluation protocol as the full model: (i) removing interaction-aware descriptors from node and edge features; (ii) disabling explicit interface modeling (thus no direct encouragement or constraint on forming cross-chain contacts); and (iii) replacing the IPA-based module with a simple multi-layer perceptron (MLP). We present ablation results, as before, across three dimensions—sequence, structure, and interface. Results are summarized in Table 3. Across all three ablations we observe a consistent, across-the-board degradation relative to the full model: lower recovery and diversity with higher perplexity, indicating a drift from the target sequence distribution and sampling collapse, reduced TM-score with increased RMSD (larger structural deviations and weakened inter-chain conformational coupling), lower lDDT, and higher $E$ and $\Delta G$ (less favorable folding and binding energetics). The success rate also drops markedly, showing that designs are less able to create or strengthen interfacial contacts.

These trends support our design hypotheses: *(a)* interaction-aware node/edge features provide critical observables that anchor interface constraints; *(b)* explicit interface modeling is necessary to align the generative objective with cross-chain contact patterns; and *(c)* the IPA-based module supplies

Table 3: Ablation Study on InterfaceDiff.

| Model | Sequence | | | Structure | | | Interface | | |
|---|---|---|---|---|---|---|---|---|---|
| | Recovery Rate %↑ | Perplexity↓ | Diversity↑ | TM-score↑ | RMSD↓ | lDDT↑ | Success Rate↑ | $E$↓ | $\Delta G$↓ |
| w/o interaction-related features | 51.46 | 4.92 | 0.31 | 0.85 | 6.36 | 0.87 | 0.58 | -36.57 | -28.98 |
| w/o interface graph | 46.24 | 5.87 | 0.26 | 0.80 | 10.36 | 0.84 | 0.37 | 23.65 | 21.71 |
| replaced IPA-based network | 20.32 | 10.56 | 0.11 | 0.52 | 14.85 | 0.73 | 0.12 | 136.54 | 125.38 |
| InterfaceDiff (Full) | **53.08** | **4.87** | **0.32** | **0.88** | **3.47** | **0.89** | **0.69** | **-53.92** | **-47.33** |

equivariant geometry and effective cross-chain information routing, coupling sequence, structure, and interface optimization. Taken together, the components are complementary and jointly enable the superior performance of the full model across all reported metrics in Table 3.

## 4 RELATED WORK

**Structure Prediction and Complex Modeling.** Deep learning has transformed protein *structure prediction* for monomers and complexes: AlphaFold2 and RoseTTAFold delivered high-accuracy predictions on monomers, while AlphaFold3 and Chai-1 expand this to multimeric complexes (Baek et al., 2021; Abramson et al., 2024; Boitreaud et al., 2024). However, such models primarily *predict* conformations for *given* sequences rather than *designing* novel sequences and structures in tandem.

**Generative Models for Sequence and Structure.** Inverse folding is a fixed-backbone generative problem for sequences: ProteinMPNN(Dauparas et al., 2022) handles both single- and multi-chain sequence design, PiFold (Gao et al., 2022) improves one-shot efficiency, and GraDe-IF (Yi et al., 2023) introduces diffusion for sequence diversity. As for structure generation, RFdiffusion (Watson et al., 2023) enables backbone and motif scaffolding; FoldingDiff (Wu et al., 2024) uses angular representations to generate plausible backbones; Chroma (Ingraham et al., 2023) provides a programmable generator for protein complexes. Nevertheless, protein–protein interface is typically handled *implicitly* as a byproduct of global modeling, offering limited control over contact patterns or interfacial energetics.

**Interface-aware Representations and Graph Methods for PPI Prediction.** Graph representations effectively model residue–residue spatial and interaction relations and are widely employed for PPI prediction. Recent graph-based PPI models include MGPPI, SE3Graph-PPI, and SpatialP-PIv2 (Zhao et al., 2024; Fang et al., 2024; Hu & Ohue, 2025). These works underscore the utility of graphs for *prediction* tasks, yet they do not *generate* new sequences or structures and usually treat the interface as emergent rather than an explicit, controllable object for design.

**Complex-related Design Tasks.** Both binder design and antibody design are tasks related to protein–protein complex design. The former designs a protein that binds to a specified protein chain(Luo et al., 2022; Watson et al., 2023; Vázquez Torres et al., 2024), whereas the latter designs the CDRs given a specified counterpart(Cutting et al., 2025; Kenlay et al., 2024). However, most current pipelines do not explicitly model the *interface* during generation, so mutation–interface coupling remains indirect.

## 5 CONCLUSION

We present InterfaceDiff, a generative model for *interface-aware* joint sequence–structure co-design of protein–protein complexes that couples intra-chain kNN graphs with an explicit bipartite interface graph and uses a lightweight IPA-inspired denoiser to sample amino-acid identities and residue-local rigid frames. INTERFACEDIFF produces diverse, high-quality, all-atom complexes; supports flexible co-design, fixed-backbone inverse folding with structure prediction, and target-conditioned binder design; and scales via sparse message passing while remaining compatible with standard packing for full-atom validation.

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

## A APPENDIX

### A.1 USE OF LLMS

We used ChatGPT (OpenAI) strictly for language editing of this manuscript. In particular, it was used to polish sentence wording and to check spelling and grammar. All suggestions were manually reviewed and integrated by the authors. The model did not generate new technical content, analyses, figures, tables, or references, and it was not used to design experiments or interpret results. No confidential data beyond the manuscript text was provided to the model. The authors take full responsibility for the paper's content.

### A.2 GRAPH CONSTRUCTION DETAILS

**Local frames and node features.** For residue $i$, we place the origin at $\mathbf{t}_i = \mathbf{C}_{\alpha,i}$ and construct a right-handed orthonormal basis from backbone bonds by Gram–Schmidt:

$$\hat{\mathbf{e}}_1 = \frac{\mathbf{C}_i - \mathbf{C}_{\alpha,i}}{\|\mathbf{C}_i - \mathbf{C}_{\alpha,i}\|}, \quad \hat{\mathbf{e}}_2 = \frac{(\mathbf{N}_i - \mathbf{C}_{\alpha,i}) - \langle \mathbf{N}_i - \mathbf{C}_{\alpha,i}, \hat{\mathbf{e}}_1 \rangle \hat{\mathbf{e}}_1}{\|(\mathbf{N}_i - \mathbf{C}_{\alpha,i}) - \langle \mathbf{N}_i - \mathbf{C}_{\alpha,i}, \hat{\mathbf{e}}_1 \rangle \hat{\mathbf{e}}_1\|}, \quad \hat{\mathbf{e}}_3 = \hat{\mathbf{e}}_1 \times \hat{\mathbf{e}}_2,$$

and set $\mathbf{R}_i = [\hat{\mathbf{e}}_1, \hat{\mathbf{e}}_2, \hat{\mathbf{e}}_3]$. Each node concatenates: a one-hot sequence identity, an eight-state secondary-structure code from a standard assignment on the complex, four coarse chemical descriptors (hydropathy, signed charge, normalized volume, polarity), trigonometric encodings of $(\phi_i, \psi_i, \omega_i)$ computed from $(\mathbf{C}_{i-1}, \mathbf{N}_i, \mathbf{C}_{\alpha,i}, \mathbf{C}_i)$, $(\mathbf{N}_i, \mathbf{C}_{\alpha,i}, \mathbf{C}_i, \mathbf{N}_{i+1})$, $(\mathbf{C}_{\alpha,i}, \mathbf{C}_i, \mathbf{N}_{i+1}, \mathbf{C}_{\alpha,i+1})$, and a multiscale exposure descriptor $\rho_i$ formed by softly aggregating neighbor displacement vectors over a geometric ladder of length scales:

$$\rho_i^{(\lambda)} = \frac{\left\| \sum_{j \in \mathcal{N}(i)} w_{ij}^{(\lambda)} (\mathbf{t}_i - \mathbf{t}_j) \right\|}{\sum_{j \in \mathcal{N}(i)} w_{ij}^{(\lambda)} \|\mathbf{t}_i - \mathbf{t}_j\|}, \qquad w_{ij}^{(\lambda)} \propto \exp\left( -\frac{\|\mathbf{t}_i - \mathbf{t}_j\|^2}{\lambda} \right).$$

For downstream all-atom reconstruction, we store per-residue heavy-atom coordinates in a fixed-size tensor with a companion mask.

**Intra-chain neighborhoods and edge attributes.** Within each chain, we build a $k$NN graph on C$\alpha$ coordinates with an adaptive $k$ that scales sublinearly with chain length and is lower bounded to ensure local connectivity. For a directed edge $(i \to j)$, we encode: (i) the clipped sequence separation $|i - j|$ via one-hot bins; (ii) a binary contact flag based on a C$\alpha$–C$\alpha$ cutoff; (iii) a radial distance embedding using Gaussian kernels with geometrically spaced bandwidths applied to the Euclidean distance; and (iv) a rigid-motion-aware orientation block

$$\mathbf{p}_{ij} = \mathbf{R}_j^\top (\mathbf{t}_i - \mathbf{t}_j), \qquad \mathbf{R}_{ij} = \mathbf{R}_j^\top \mathbf{R}_i,$$

where $\mathbf{p}_{ij}$ is the relative displacement expressed in the destination frame and $\mathbf{R}_{ij}$ is the relative rotation, which we vectorize for learning.

**Interface bipartite graph and cross-chain features.** To extract the interaction surface, we select the two longest polypeptide chains and identify cross-chain residue pairs by a heavy-atom radius search; for each residue in the longer chain, all residues in the partner chain that contain any non-hydrogen atom within a short Å-scale radius are considered interfacial partners, and we instantiate a directed edge for each detected pair. Interface edges reuse the geometric core above (distance embedding plus $\mathbf{p}_{ij}$ and $\mathbf{R}_{ij}$ computed with the joint $(\mathbf{R}, \mathbf{t})$ of both chains) and append residue–residue compatibility features derived from node-chemistry (polarity match, opposite-charge indicator, absolute differences in normalized volume and hydropathy), together with four canonical motif detectors based on explicit distance tests over residue-specific atom sets: hydrogen bonds (donor–acceptor within a short cutoff including backbone N/O), salt bridges (acidic–basic side-chain atoms within a slightly larger cutoff), aromatic–aromatic interactions (ring-centroid proximity for PHE/TYR/TRP), and cation–$\pi$ interactions (charged nitrogens to aromatic centroids). If no cross-chain neighbor is found the complex is treated as non-interacting and skipped in training; on curated heterodimers this condition is rare. The per-chain graphs and the interface bipartite graph are finally packaged with $(\mathbf{R}, \mathbf{t})$ and heavy-atom tensors to support sparse message passing both within chains and across the interface, aligning with interface-aware co-design objectives.

A.3 EXPONENTIAL MAP ON SO(3)

We parametrize 3D rotations by a *rotation vector* $\mathbf{v}_i \in \mathbb{R}^3$ and map it to a rotation matrix $\mathbf{R}_i \in$ SO(3) via the matrix exponential

$$\mathbf{R}_i = \exp([\mathbf{v}_i]_\times), \qquad [\mathbf{v}]_\times = \begin{bmatrix} 0 & -v_z & v_y \\ v_z & 0 & -v_x \\ -v_y & v_x & 0 \end{bmatrix}, \tag{12}$$

where $[\mathbf{v}]_\times$ is the skew-symmetric ("hat") operator satisfying $[\mathbf{v}]_\times \mathbf{w} = \mathbf{v} \times \mathbf{w}$ for any $\mathbf{w} \in \mathbb{R}^3$. Geometrically, letting $\theta = \|\mathbf{v}\|$ and $\mathbf{u} = \mathbf{v}/\theta$ (if $\theta \neq 0$), $\mathbf{R} = \exp([\mathbf{v}]_\times)$ represents a right-handed rotation by angle $\theta$ about axis $\mathbf{u}$.

**Closed form (Rodrigues).** The exponential admits the Rodrigues formula

$$\mathbf{R} = \mathbf{I} + A(\theta) [\mathbf{v}]_\times + B(\theta) [\mathbf{v}]_\times^2, \qquad A(\theta) = \frac{\sin\theta}{\theta}, \ B(\theta) = \frac{1 - \cos\theta}{\theta^2}, \tag{13}$$

with $\mathbf{I}$ the $3 \times 3$ identity. For small $\theta$, we use the series

$$A(\theta) \approx 1 - \frac{\theta^2}{6} + \frac{\theta^4}{120}, \qquad B(\theta) \approx \frac{1}{2} - \frac{\theta^2}{24} + \frac{\theta^4}{720}, \tag{14}$$

which yields numerically stable evaluations of equation 13.

**Differentials and backpropagation.** The differential of the exponential map can be written using the *left Jacobian* $\mathbf{J}_l(\mathbf{v}) \in \mathbb{R}^{3 \times 3}$:

$$d\mathbf{R} = \mathbf{R} \left[ \mathbf{J}_l(\mathbf{v}) \, d\mathbf{v} \right]_\times, \qquad \mathbf{J}_l(\mathbf{v}) = \mathbf{I} - \frac{1 - \cos\theta}{\theta^2} [\mathbf{v}]_\times + \frac{\theta - \sin\theta}{\theta^3} [\mathbf{v}]_\times^2. \tag{15}$$

Equation equation 15 is convenient for computing gradients of losses defined on $SO(3)$ with respect to the vector parameter $\mathbf{v}$.

**Log map and invertibility.** The inverse map (up to the usual $\pi$-angle axis sign ambiguity) is

$$\theta = \arccos\left( \frac{\text{tr}(\mathbf{R}) - 1}{2} \right), \qquad \mathbf{v} = \frac{\theta}{2 \sin\theta} (\mathbf{R} - \mathbf{R}^\top)^\vee, \tag{16}$$

where $(\cdot)^\vee$ is the inverse of the hat operator. In practice we clamp the argument of $\arccos$ to $[-1, 1]$, use the series in equation 14 for $\theta \approx 0$, and handle $\theta \approx \pi$ with care due to axis ambiguity. These choices make equation 12–equation 16 robust in training and inference.

### A.4  BACKGROUND ON DENOISING DIFFUSION PROBABILISTIC MODELS

Denoising Diffusion Probabilistic Models (DDPMs) (Ho et al., 2020; Nichol & Dhariwal, 2021) are a class of latent variable models designed for high-fidelity data generation. They conceptualize generation as the reversal of a fixed data corruption process. This is accomplished through two complementary Markov processes: a **forward (diffusion) process** that gradually adds noise to data, and a learned **reverse (generative) process** that systematically removes it.

#### A.4.1  FORWARD PROCESS

The forward process, denoted by $q$, incrementally perturbs a clean data sample $x_0 \sim q(x_0)$ over $T$ discrete time steps by adding Gaussian noise. The process is governed by a predefined variance schedule $\{\beta_t\}_{t=1}^T$, where $\beta_t \in (0, 1)$. The distribution of the noisy sample $x_t$ at step $t$ given the sample from the previous step $x_{t-1}$ is defined as:

$$q(x_t | x_{t-1}) = \mathcal{N}(x_t; \sqrt{1 - \beta_t} x_{t-1}, \beta_t \mathbf{I}) \tag{17}$$

where $\mathcal{N}(\cdot; \mu, \sigma^2 \mathbf{I})$ is a Gaussian distribution with mean $\mu$ and covariance $\sigma^2 \mathbf{I}$.

A significant property of this process is the ability to sample $x_t$ at any arbitrary timestep $t$ directly from the original data $x_0$. By setting $\alpha_t = 1 - \beta_t$ and $\bar{\alpha}_t = \prod_{s=1}^t \alpha_s$, the conditional distribution $q(x_t | x_0)$ can be expressed in a closed form:

$$q(x_t | x_0) = \mathcal{N}(x_t; \sqrt{\bar{\alpha}_t} x_0, (1 - \bar{\alpha}_t) \mathbf{I}) \tag{18}$$

This allows for efficient training, as we can reformulate $x_t$ using the reparameterization trick with a standard Gaussian noise variable $\epsilon \sim \mathcal{N}(\mathbf{0}, \mathbf{I})$:

$$x_t = \sqrt{\bar{\alpha}_t} x_0 + \sqrt{1 - \bar{\alpha}_t} \epsilon \tag{19}$$

As $t \to T$, the distribution $q(x_T | x_0)$ converges to an isotropic Gaussian distribution, $p(x_T) \approx \mathcal{N}(\mathbf{0}, \mathbf{I})$, effectively erasing all information from the original sample $x_0$.

#### A.4.2  REVERSE PROCESS

The generative part of the model is the reverse process, $p_\theta$, which aims to reverse the forward diffusion. It starts with a sample $x_T$ drawn from the prior, $p(x_T) = \mathcal{N}(\mathbf{0}, \mathbf{I})$, and learns to iteratively denoise it to produce a sample $x_0$ that resembles the true data distribution. Each transition in this chain is defined as a conditional Gaussian parameterized by a neural network:

$$p_\theta(x_{t-1} | x_t) = \mathcal{N}(x_{t-1}; \mu_\theta(x_t, t), \Sigma_\theta(x_t, t)) \tag{20}$$

While the exact posterior $q(x_{t-1} | x_t, x_0)$ is tractable, it requires knowledge of $x_0$, which is unavailable during generation. Therefore, the model approximates this posterior. Instead of predicting the

mean $\mu_\theta$ directly, it is common practice to train a neural network, $\epsilon_\theta(x_t, t)$, to predict the noise component $\epsilon$ from Equation 19. The mean of the reverse transition is then parameterized as:

$$\mu_\theta(x_t, t) = \frac{1}{\sqrt{\alpha_t}} \left( x_t - \frac{\beta_t}{\sqrt{1 - \bar{\alpha}_t}} \epsilon_\theta(x_t, t) \right) \tag{21}$$

The variance $\Sigma_\theta(x_t, t)$ is typically set to a non-learned, time-dependent constant, such as $\Sigma_\theta(x_t, t) = \beta_t \mathbf{I}$.

The model is trained by optimizing a simplified objective function that minimizes the mean squared error between the true and predicted noise at each step:

$$L_{\text{simple}}(\theta) = \mathbb{E}_{t \sim [1,T], x_0, \epsilon} \left[ \left\| \epsilon - \epsilon_\theta(\sqrt{\bar{\alpha}_t} x_0 + \sqrt{1 - \bar{\alpha}_t}\epsilon, t) \right\|^2 \right] \tag{22}$$

This objective directly trains the network to predict and remove the noise added during the forward process, enabling the reverse process to generate realistic data samples.

### A.5 DETAILED ARCHITECTURE OF THE IPA-BASED DENOISER NETWORK

This appendix provides the technical details for the graph denoiser network described in the main text. The network updates node hidden representations $h_i$ for each residue $i$ by aggregating information from its neighbors $j \in \mathcal{N}(i)$ using a geometric attention mechanism.

**Geometric Attention.** The attention weight $\alpha_{ij}$ between two nodes is computed from a combination of three distinct signals:

$$\alpha_{ij} = \underset{j \in \mathcal{N}(i)}{\text{Softmax}} \left( \frac{1}{\sqrt{3}} (\text{logits}_{\text{node}} + \text{logits}_{\text{pair}} + \text{logits}_{\text{spatial}}) \right)$$

The components are defined as follows:

- **logits$_{\text{node}}$**: A standard attention score based on feature similarity. It is the dot product between query and key vectors that are linearly projected from the node features $h_i$ and $h_j$.
- **logits$_{\text{pair}}$**: A bias term derived directly from the graph structure. It is computed by a linear projection of the edge attributes $E_{ij}$ connecting the two residues.
- **logits$_{\text{spatial}}$**: An SE(3)-invariant geometric score based on the distances between sets of virtual "attention points" attached to each residue's local frame. It is formulated as:

$$\text{logits}_{\text{spatial},ij} = -\frac{\gamma}{2} \sqrt{\frac{2}{9P_q}} \sum_{p=1}^{P_q} \|(x_{t,i}^{\text{rot}} \mathbf{q}_p^i + x_{t,i}^{\text{trans}}) - (x_{t,j}^{\text{rot}} \mathbf{k}_p^j + x_{t,j}^{\text{trans}})\|^2$$

where $(x_t^{\text{rot}}, x_t^{\text{trans}})$ are the residue's noisy frame coordinates at timestep $t$. The local coordinates for the query points $\mathbf{q}_p^i \in \mathbb{R}^3$ and key points $\mathbf{k}_p^j \in \mathbb{R}^3$ are learned via linear projections from the node features $h_i$ and $h_j$, respectively. $P_q$ is the number of query/key points, and $\gamma$ is a learnable scaling parameter.

**Information Aggregation and Update.** Using the computed attention weights $\alpha_{ij}$, the block aggregates information along three pathways: (1) a weighted sum of value features projected from neighboring node representations $h_j$; (2) a weighted sum of the edge features $E_{ij}$; and (3) a geometric pathway where a set of "value points", learned similarly to query/key points, are aggregated. The resulting local coordinates, directions, and norms of these aggregated value points serve as geometric features. These three streams of information are concatenated, passed through a feed-forward network, and used to update the node representation $h_i$ via a residual connection.

**Decoder Input.** After the final attention block, the resulting node representations $h_i^{\text{final}}$ are conditioned on the diffusion timestep $t$. A time embedding is created from the corresponding noise schedule parameter $\beta_t$ and concatenated with the node features to form the input for the final decoders:

$$\mathbf{in}_i = \text{Concat}(h_i^{\text{final}}, [\beta_t, \sin(\beta_t), \cos(\beta_t)])$$

This combined vector $\mathbf{in}_i$ is then processed by three separate MLP decoders to predict $\widehat{\mathbf{c}}_{0,i}$, $\widehat{\mathbf{R}}_{0,i}$ (via its axis-angle vector), and $\widehat{\epsilon}_{t,i}^{\text{local}}$.

## A.6 DATASET PREPARATION

To construct a high-quality dataset of protein-protein complexes, we employed a multi-step filtering process. To ensure the reliability and completeness of the data, we curated authentic protein complexes from the latest release of the PDBBurley et al. (2025), PDBbind v2020, and PINDERKovtun et al. (2024) for model training and evaluation. First, we selected protein-protein complexes with structural resolution better than 3.5 Å, and exclude complexes containing RNA or DNA structures.

Specifically, we focused exclusively on heterodimers for two key reasons. First, heterodimers typically exhibit more complex interaction interfaces characterized by asymmetric geometric, sequence, and physicochemical features, making them ideal for studying diverse and intricate cross-chain interactions. Second, homodimers often involve symmetric chains with repetitive features, which can lead to overfitting in machine learning models and hinder their ability to generalize to more nuanced heterodimeric interactions.

To ensure the suitability of the dataset for downstream modeling tasks, we filtered out proteins with chain lengths shorter than 30 residues. Chains shorter than this threshold may lack sufficient structural complexity and functional relevance to contribute meaningfully to the analysis of protein-protein interactions. Additionally, proteins with chain lengths exceeding 512 residues were excluded to maintain computational efficiency. Finally, to guarantee data quality, the PDBfixer tool was used to repair protein structures, ensuring that all atomic coordinates for amino acid residues were complete. This comprehensive filtering process resulted in a curated dataset of protein-protein complexes suitable for downstream analysis and modeling.

## A.7 EVALUATION METRICS — DEFINITIONS AND FORMULAS

### A.7.1 SEQUENCE-LEVEL METRICS

**Sequence Recovery** Let the sequence length be $N$. Denote the one-hot encoding of the native amino acid at position $i$ by $\mathbf{y}_i \in \{0,1\}^{20}$ and the model prediction (probability or one-hot) by $\hat{\mathbf{y}}_i \in [0,1]^{20}$. With $\arg\max$ mapping to a discrete residue identity, the recovery is

$$\text{Rec} = \frac{1}{N}\sum_{i=1}^{N}\mathbf{1}\Big(\arg\max \mathbf{y}_i = \arg\max \hat{\mathbf{y}}_i\Big). \tag{23}$$

**Perplexity** Let $\mathbf{p}_i \in [0,1]^{20}$ be the predicted categorical distribution at position $i$, and $\mathbf{y}_i$ the one-hot ground truth. The mean cross-entropy is

$$\text{CE} = -\frac{1}{N}\sum_{i=1}^{N}\sum_{a=1}^{20} y_{i,a}\log p_{i,a} = -\frac{1}{N}\sum_{i=1}^{N}\log p_{i,a^\star(i)}, \tag{24}$$

where $a^\star(i)$ is the native residue index at position $i$. Perplexity is

$$\text{PPL} = \exp\big(\text{CE}\big). \tag{25}$$

**Sequence Diversity** For the same target, suppose $M$ sampled sequences are generated, each with discrete residues $s_i^{(m)}$ at position $i$ for sample $m$. The average pairwise similarity across all unordered pairs $(m,n)$, $m < n$, is

$$\text{Sim} = \frac{2}{M(M-1)}\sum_{1\le m<n\le M}\left(\frac{1}{N}\sum_{i=1}^{N}\mathbf{1}\big(s_i^{(m)} = s_i^{(n)}\big)\right). \tag{26}$$

Diversity is then defined as

$$\text{Div} = 1 - \text{Sim}. \tag{27}$$

### A.7.2 STRUCTURE-LEVEL METRICS

**Ramachandran plot** report the percentages of residues whose $(\phi, \psi)$ dihedrals fall in favored regions. **Backbone RMSD** is computed after optimal superposition (e.g., Kabsch) using main-chain

atoms (N, $C_\alpha$, C):

$$\text{bbRMSD} = \sqrt{\frac{1}{N_\text{bb}} \sum_{i=1}^{N_\text{bb}} \left\| \tilde{\mathbf{r}}_i - \tilde{\mathbf{r}}_i' \right\|^2}, \tag{28}$$

where $\tilde{\mathbf{r}}_i$ and $\tilde{\mathbf{r}}_i'$ denote optimally superposed backbone atom coordinates.

The **TM-score** (Template Modeling score)Zhang & Skolnick (2004) is a length-independent metric for measuring the structural similarity between two protein structures. Unlike RMSD, which can be dominated by local deviations, TM-score normalizes by protein length and applies a distance-dependent weighting function. The score ranges from 0 (no similarity) to 1 (perfect match), with values above 0.5 generally indicating the same fold and values below 0.2 corresponding to random similarity.

$$\text{TM-score} = \max \left[ \frac{1}{L_\text{target}} \sum_{i=1}^{L_\text{ali}} \frac{1}{1 + \left( \frac{d_i}{d_0(L_\text{target})} \right)^2} \right] \tag{29}$$

where:

- $L_\text{target}$: length of the target protein.
- $L_\text{ali}$: number of aligned residue pairs.
- $d_i$: distance between the $i$-th pair of aligned C$\alpha$ atoms.
- $d_0(L_\text{target})$: scale parameter depending on protein length, defined as

$$d_0(L_\text{target}) = 1.24 \sqrt[3]{L_\text{target} - 15} - 1.8$$

This formulation makes TM-score less sensitive to local structural variations and more reliable for assessing overall fold similarity. Scores above **0.5** typically indicate the same fold, while scores below **0.2** correspond to random structural similarity.

**lDDT** The Local Distance Difference Test (lDDT) is a *superposition-free* local structural similarity metric that evaluates how well a predicted structure preserves *pairwise distances* relative to a reference (Mariani et al., 2013). Unlike global metrics such as RMSD, lDDT is insensitive to rigid-body motions and domain reorientations, focusing instead on the geometric consistency of each residue (or atom) within its local environment. This yields robustness on multi-domain proteins, flexible loops, and protein–protein interfaces.

Let the reference and predicted structures be $\{\mathbf{x}_i^\star\}_{i=1}^M$ and $\{\mathbf{x}_i\}_{i=1}^M$ (x refers to atoms or residues). For each center $i$, define the reference neighborhood $\mathcal{N}_i = \{j \neq i : \|\mathbf{x}_i^\star - \mathbf{x}_j^\star\| \leq R_0\}$, with neighborhood radius $R_0$ (commonly $R_0 \in [10, 15]$ Å). Given distance tolerances $\mathcal{T} = \{\tau_1, \ldots, \tau_K\}$ (commonly $\{0.5, 1, 2, 4\}$ Å), define

$$s_{ij} = \frac{1}{K} \sum_{k=1}^K \mathbb{1} \left( \left| \|\mathbf{x}_i - \mathbf{x}_j\| - \|\mathbf{x}_i^\star - \mathbf{x}_j^\star\| \right| < \tau_k \right).$$

The per-center score and global lDDT are

$$\text{lDDT}_i = \frac{1}{|\mathcal{N}_i|} \sum_{j \in \mathcal{N}_i} s_{ij}, \qquad \text{lDDT} = \frac{1}{M} \sum_{i=1}^M \text{lDDT}_i,$$

taking values in $[0, 1]$ (often reported on $[0, 100]$). Neighborhoods are defined on the *reference* structure and empty neighborhoods are typically excluded in practice.

There are some variants of lDDT: (i) *lDDT-C$\alpha$*: builds neighborhoods and distances on C$\alpha$ only; fast and sensitive to backbone geometry. (ii) *All-atom lDDT*: captures side-chain fidelity but requires consistent handling of missing atoms and altLocs. (iii) *Region-/interface-restricted lDDT*: computed on subsets such as binding sites or PPI interfaces (e.g., residues with any heavy atom within 8 Å across chains), reported as lDDT-BS / interface-lDDT.

RMSD is affected by alignment and outliers; TM-score emphasizes global topology. lDDT offers a *local* perspective on geometric fidelity, making it particularly informative for multi-domain motion, flexible loops, active sites, and interfaces. In reporting, we recommend combining global metrics (TM-score/RMSD) with lDDT or region-restricted lDDT.

The **MolProbity score**Chen et al. (2010) is a widely used structural quality metric for macromolecular models. It combines several geometric validation measures—particularly *clashscore*, *rotamer outliers*, and *Ramachandran outliers*—into a single number that is on the same scale as crystallographic resolution (Å). A lower score indicates a model of higher quality. **Clashscore** is number of all-atom steric overlaps greater than 0.4 Å per 1000 atoms, rotamer outliers is percentage of residues with side-chain conformations outside favored rotamers, and **Ramachandran outliers** is the percentage of residues falling outside the favored/allowed regions of the Ramachandran plot. This composite scoring system allows for intuitive interpretation: structures with **MolProbity scores close to their experimental resolution** (e.g., $\sim 2.0$ for a 2 Å structure) are considered very good, while higher scores suggest poorer stereochemical quality.

### A.7.3 Interface-level Metrics

**Success Rate**    Let $C_{\mathrm{nat}}$ be the native interface contact count, computed with the same atom selection and distance threshold as used for designed structures (heavy-atom pairs within $\tau = 6$ Å). For the $k$-th design of the same target, let $C^{(k)}$ be its contact count. Over $K$ designs,

$$\mathrm{SR} \;=\; \frac{1}{K}\sum_{k=1}^{K}\mathbf{1}\big(C^{(k)} > C_{\mathrm{nat}}\big),\tag{30}$$

and we aggregate across targets by averaging target-level SRs (we report the chosen aggregation in the main text).

**Total Energy** $E$    We compute the total energy $E(x)$ of a given conformation $x$ using PyRosetta's scoring function(Chaudhury et al., 2010).

**Binding energy** $\Delta G$.    Given a complex conformation $x_{AB}$, we approximate the binding free energy as

$$\Delta G = E(x_{AB}) - \big(E(x_A^*) + E(x_B^*)\big),$$

where $x_A^*$ and $x_B^*$ denote monomer states obtained by spatially separating the partners (retaining their internal conformations) and then repacking side chains and locally minimizing around the interface under the same scoring function. This is equivalent to Rosetta's standard *interface* $\Delta G$ protocol: evaluate the complex energy, then evaluate the separated partners in an "environment-matched" setting, and take the difference. Negative $\Delta G$ indicates favorable binding under this energy function.

