# OpenReview forum: "InterfaceDiff: Interface-Aware Sequence-Structure Co-Design of Protein Complexes with Graph-Based Diffusion"
_ICLR.cc/2026/Conference — ICLR 2026 Conference Withdrawn Submission_

### Official Review · Reviewer_QoPu · 2025-10-28

**Soundness:** 2
**Presentation:** 3
**Contribution:** 2
**Rating:** 2
**Confidence:** 4

**Summary:**

The paper proposes InterfaceDiff, a graph-based diffusion framework for interface-aware sequence–structure co-design of protein complexes. The method represents each chain as a residue-level graph, models the interface via an explicit bipartite graph, and leverages an IPA-inspired denoiser to jointly generate amino-acid sequences and local rigid frames, followed by side-chain packing with existing tools.

**Strengths:**

1. The authors attempt to address an interface-aware protein–complex design problem, which is an important aspect for practical applications in binder and therapeutic development.

2. The explicit bipartite interface graph is conceptually clear and may offer fine-grained control over cross-chain contacts.

3. Ablation experiments are included to show the contributions of individual components.

**Weaknesses:**

1. Limited Novelty: Much of the Method section reiterates standard techniques, including established graph construction schemes and discrete–continuous diffusion setups. The denoiser is a direct adaptation of Invariant Point Attention, and side-chain packing relies on existing tools. Overall, the methodological contribution is incremental.

2. Feasibility Concerns: The denoiser conditions on detailed intra-/inter-chain graphs; however, the inter-chain graph is sparse and requires precise contact data. In real complex-design scenarios such information may be unavailable(typically, only hotspot-level cues are available, similar to RFDiffusion), raising concerns about applicability in practice and possible **label leakage** in evaluation.

3. Experimental Setting Issues: Baselines are mostly inverse folding models, which are ill-suited for full co-design. In Sec. 3.1 this makes comparisons questionable; in Sec. 3.2 InterfaceDiff-bb trails Bridge-IF on several key metrics; in Sec. 3.3 the binder design task omits strong SOTA baselines such as RFdiffusion, AlphaProteo, and PXDesign, limiting the evidence for superiority.

4. Incomplete Metrics: Structural diversity, an important facet of design quality, is not reported in the evaluation.

**Questions:**

Please refer to the Weaknesses section. If the concerns in W2 and W3 are satisfactorily addressed, my overall assessment could improve significantly.

---

### Official Review · Reviewer_1sof · 2025-10-29

**Soundness:** 3
**Presentation:** 2
**Contribution:** 2
**Rating:** 4
**Confidence:** 4

**Summary:**

The paper introduces InterfaceDiff, a graph-based diffusion framework for the joint sequence-structure co-design of protein-protein complexes (PPCs). The authors argue that existing methods often fail to treat the functionally critical protein-protein interface as a primary design target. The core novelty is its representation: it encodes the complex using intra-chain graphs coupled by an explicit bipartite interface graph. This graph connects interacting residues across chains and is enriched with geometric and chemical features. The model learns a joint distribution over discrete amino acid sequences and continuous local rigid frames (rotations $SO(3)$ and translations $\mathbb{R}^{3}$) via a simultaneous denoising process. This is powered by a computationally efficient, IPA-inspired graph neural network denoiser that operates on sparse graphs and local frames. The framework is evaluated on co-design, fixed-backbone design, and binder design tasks7

**Strengths:**

1. The paper correctly identifies the main challenge in PPC design: treating the interface as a "first-class target". Its core representation, an explicit bipartite interface graph, directly addresses this by concentrating modeling capacity on physically interacting residues. This explicit interface graph is a significant novelty and is enriched with specific "interaction semantics" (like hydrogen bonds, salt bridges, and hydrophobics), providing chemically-aware control levers that are missing from implicit, all-to-all attention models.

2. The paper makes smart design choices by using an IPA-inspired denoiser operating on sparse graphs and residue-local frames. This captures essential 3D cues and handles rotations on SO(3) in a standard, well-specified way while avoiding the high computational overhead of fully SE(3)-equivariant networks, thus improving scalability.

3. The model demonstrates top-tier results on the primary sequence-structure co-design task (Table 1). It achieves the highest sequence recovery (53.08%) and best perplexity (4.87) among baselines, all while simultaneously producing the best structural metrics (TM-score 0.88) and the most physically plausible interfaces (lowest ΔG, highest Success Rate). It also excels at the practical task of binder design (Table 2), outperforming all baselines across sequence, structure, and interface quality.

4. The paper's structural claims are well-supported by a comprehensive evaluation pipeline. It doesn't just stop at backbone metrics but reconstructs full all-atom models, which are then validated using standard biophysical tools like MolProbity and Rosetta energy scoring.

5. Finally, the ablation experiments (Table 3) provide exceptionally strong, causal support for the paper's design. Removing either the explicit interface graph or the IPA-based geometric module causes a dramatic and measurable collapse in performance, directly proving that each new component is essential to the model's success.

**Weaknesses:**

While the method is promising, the paper has several significant weaknesses that should be addressed before acceptance.

First, the primary interface metric, "Success Rate" (SR), is poorly defined and potentially misleading. The paper defines success as any design with more heavy-atom contacts than the native structure ($C^{(k)} > C_{nat}$). This metric risks "reward hacking," as it would positively score overly "sticky" or non-specific designs while penalizing a high-fidelity design that might be sparser but correct. A more robust metric, such as a contact-map F1-score relative to the native interface or an interface-specific lDDT score, would be far more informative.

Second, the experimental comparison to baselines is not "apples-to-apples." For sequence-based baselines (e.g., ProteinMPNN), structures are generated by feeding the designed sequences into an external predictor (Chai-1). InterfaceDiff, however, generates its own structures. This makes it impossible to disentangle whether InterfaceDiff is a better sequence generator or simply a better structure generator. A fair comparison would require either adding co-design structural baselines (like RFdiffusion) or running a symmetric pipeline where InterfaceDiff's sequences are also folded by Chai-1.

Third, the paper completely lacks computational transparency, which undermines claims of scalability and efficiency. Key details required for reproducibility are missing, including: model parameter counts, the total number of diffusion steps ($T$), training steps/epochs, batch sizes, and the hardware used (GPU type, VRAM, and wall-clock training time).

Fourth, several key methodological details are vague. The model's core component—the interface graph—is built using a "threshold 7" (presumably a typo for $\tau$), but the exact value is not specified in the methods. It is unclear if it is the same $\tau=6~A$ used for the SR metric. The model's performance could be highly sensitive to this choice, yet no robustness analysis is provided. Similarly, the paper claims an advantage by using an invariant IPA-inspired block, but the trade-offs versus a full equivariant network are under-analyzed. A simple test on randomly-oriented complexes would be needed to validate this design choice.

Finally, the paper's claims about scope are broader than what the evidence supports. The training data was "exclusively on heterodimers", so any applicability to homodimers, larger stoichiometries, or symmetric systems is purely speculative. Furthermore, the abstract and contributions list "antibody design" as an application, but no specific CDR design benchmarks are presented, only a general binder design task. This claim should be substantiated with relevant experiments or softened.

**Questions:**

1. Could you provide the missing computational details required for reproducibility, including the model's parameter count, the total number of diffusion steps ($T$), the training hardware (GPU type, VRAM), and the total training time and batch size?

2. Can you justify the "Success Rate" (SR) metric, which rewards any contact count higher than native ($C^{(k)} > C_{nat}$), and provide results using a more standard metric like native contact F1-score or interface-lDDT to validate interface quality?

3. Sequence-only baselines are folded with Chai-1, while your method generates structures directly. Please (a) add structure-generation baselines for complexes or (b) also fold your designed sequences with the same folder (Chai-1) and compare on the same metrics, so the pipeline is symmetric.

4. What was the specific distance threshold ($\tau$) used for constructing the interface graph during training, and can you provide a sensitivity analysis on how this critical hyperparameter affects performance?

5. Given that the experiments only show a general binder design task, can you provide specific CDR loop design benchmarks to support the paper's claim of being applicable to "antibody design," or would you be willing to soften this claim?

6. Will you be releasing the exact training/validation/test set splits and the full model and evaluation code (including PyRosetta/FlowPacker scripts) to ensure the results are fully reproducible?

---

### Official Review · Reviewer_PvXE · 2025-10-31

**Soundness:** 1
**Presentation:** 3
**Contribution:** 1
**Rating:** 2
**Confidence:** 4

**Summary:**

This work proposes explicitly modeling the interactions between two proteins in a complex, viewed in terms of their graph representations, via an explicit sparse interface determined based on inter-atomic proximity (rather than considering all pairwise interactions). A local frame-based representation is employed, and a diffusion process is set up over joint distribution of residues and continuous local (frame-wise) transformations (namely, rotation and translation) conditioned on the residual-level protein graphs and their (sparse) interface graph. The interface graph is subjected to message passing with a GNN denoiser based on invariant point attention. A post-processing step accounts for the side chains to reconstruct the complex. Empirical evaluation is performed with datasets derived from PDB, PDBbind v2020, and PINDER.

**Strengths:**

--- Sequence-structure co-design of protein complexes is an important topic, and an active area of research within ML for drug discovery.

--- The paper is generally well written and should be easy-to-read for a broad audience.

**Weaknesses:**

--- I have significant concerns about novelty of this work. For examples, a prominent earlier work on protein sequence design conditioned on graph structure [1] already proposed local frames and sparse k-NN based graph representations (and considered both rigid and flexible backbone settings).

--- Apriori, there is no reason why the model with full message passing cannot learn sparse connections. Furthermore, one could use e.g., sparsity promoting regularizers to induce sparsity if needed.

--- It's unclear how the proposed method performs on tasks such as antibody structure and sequence co-design for which strong methods are known; e.g., [2, 3].

--- Some other important earlier works that model interactions between proteins/molecules using inter-graph (hierarchical) message passing have also not been referenced, discussed, or compared with; e.g., [4, 5, 6].

--- Based on the description in section 2.2, it seems that entire graph structure (residue level graphs as well as the interaction interface) is fixed (upto local rotations and translations) and only the node-level (residue) attributes are noised and denoised during the diffusion process. While this is fine for applications such as binding where conditioning on structure seems to suffice [7], I'm concerned it would not work for problems such as de novo antibody structure and sequence co-design where we typically do not know the structure of antibody in advance.

--- The behavior/performance of the proposed method for pairs of proteins with different lengths and interface sizes is unclear.

--- No standard errors have been reported in experiments, which raises concerns about statistical soundness of evaluation.


[1] Ingraham et al. Generative models for graph-based protein design. NeurIPS 2019.

[2] Shi et al. Protein Sequence and Structure Co-Design with Equivariant Translation. ICLR 2023.

[3] Verma et al. AbODE: Ab initio Antibody Design using Conjoined ODEs. ICML 2023.

[4] Start, Ganea et al. EquiBind: Geometric Deep Learning for Drug Binding Structure Prediction, ICML 2022.

[5] Ganea et al. Independent SE(3)-Equivariant Models for End-to-End Rigid Protein Docking. ICLR 2022.

[6] Jin et al. Antibody-antigen Docking and Design via Hierarchical Equivariant Refinement. ICML 2022.

[7] Karczewski et al. What Ails Generative Structure-based Drug Design: Too Little or Too Much Expressivity. AISTATS 2025.

**Questions:**

Could you please address all the concerns I outlined under the weaknesses section? I would be willing to revisit my score if my concerns are satisfactorily addressed, and in particular, if you could demosntrate clear empirical benefits of your method over the approaches from references [2, 3] therein on antibody structure and sequence co-design.

---

### Official Review · Reviewer_zMYU · 2025-11-01

**Soundness:** 2
**Presentation:** 1
**Contribution:** 2
**Rating:** 2
**Confidence:** 5

**Summary:**

The paper proposed InterfaceDiff to design protein-protein complexes with aware interface. It models the intra-chain graph and inter-chain interface graph, and applied diffusion model to achieve design. The proposed method is evaluated on the general protein-protein complex design and target protein binder complex design tasks.

**Strengths:**

The idea of co-designing protein-protein complex sequence and structure is interesting and useful.

**Weaknesses:**

1. The problem setting of this paper has some issues: The paper assume we already know the protein-protein complex interface, based on which it models the intra-chain graph and inter-chain interface graph. However, in reality, the interface is unknown. Basically, what we know might only be the target protein. If we already know the interface, the complex design task would not be challenging any more. The assumption of this paper is not practical in reality.

2. The experimental setting also has some issues: In table1 and table2, almost all the baselines are inverse folding models, like ProteinMPNN, PiFold. Most of these models are neither sequence-structure co-design models nor protein-protein complex design models. The comparisons are totally unfair and meaning less.

3. The metrics are not indicative enough. For co-design parts, there should be some consistency metrics between the designed sequence and structure. For the complex design parts, there should be something like binding affinity scores or AlphaFold3 ipTM scores. Metrics like the sequence recovery rate is not useful at all.

**Questions:**

Please see above weaknesses.

---

### Note · Authors · 2025-11-13

I have read and agree with the venue's withdrawal policy on behalf of myself and my co-authors.